# AI-Augmented Advances in the Diagnostic Approaches to Endometrial Cancer

**DOI:** 10.3390/cancers17111810

**Published:** 2025-05-28

**Authors:** Nabiha Midhat Ansari, Usman Khalid, Daniel Markov, Kristian Bechev, Vladimir Aleksiev, Galabin Markov, Elena Poryazova

**Affiliations:** 1Faculty of Medicine, Medical University of Plovdiv, 4002 Plovdiv, Bulgaria; usmankhalid957@gmail.com (U.K.); gabi_markov@abv.bg (G.M.); 2Department of General and Clinical Pathology, Medical University of Plovdiv, 4002 Plovdiv, Bulgaria; daniel.markov@mu-plovdiv.bg (D.M.); kristian_bechev@abv.bg (K.B.); eporiazova@abv.bg (E.P.); 3Department of Clinical Pathology, University Multidisciplinary Hospital for Active Treatment “Pulmed”, 4002 Plovdiv, Bulgaria; 4Neurological Surgery, Pulmed University Hospital, 4000 Plovdiv, Bulgaria; 5Department of Thoracic Surgery, UMHAT “Kaspela”, 4002 Plovdiv, Bulgaria; vl_alex@abv.bg; 6Department of Cardiovascular Surgery, Medical University of Plovdiv, 4002 Plovdiv, Bulgaria

**Keywords:** artificial intelligence, endometrial cancer, diagnostic imaging, histology, machine learning, multi-omics, deep learning, gynecologic oncology

## Abstract

Endometrial cancer is the most common cancer of the female reproductive system in developed countries. Detecting it early and accurately is important for successful treatment and better outcomes. New technologies using artificial intelligence (AI) are showing promise in helping doctors diagnose this cancer more precisely and quickly. This review looks at how AI is being used in different ways, such as analyzing tissue samples, medical images, and complex biological data, to improve the diagnosis of endometrial cancer. The goal is to understand how these new tools work, how they might be used in hospitals, and what challenges still need to be solved. By exploring current research in this area, we hope to highlight the potential of AI to support doctors in making better decisions and to encourage further studies that can bring these tools into everyday medical practice.

## 1. Introduction

Endometrial cancer is one of the most common tumors in women, with a higher prevalence in Europe and North America [1]. Data from 1990 to 2021 reveal a significant global rise in endometrial cancer cases among individuals aged 55 and older. Over this 30-year period, the number of cases more than doubled, increasing from 141,173 in 1990 to 360,253 in 2021. Correspondingly, the incidence rate per 100,000 population rose from 39.22 to 45.81, highlighting a growing public health concern [2]. Over 417,000 women worldwide were diagnosed with EC in 2020, representing a 132% increase over the past 30 years [3]. EC cases are anticipated to grow by about 40% between 2020 and 2040. Each year in Europe, around 121,000 women are expected to receive a diagnosis of either recurrent or primary advanced EC, and, at the time of diagnosis, about 15–20% of patients would already have the advanced-stage disease [4]. The country cancer profile for Bulgaria found that, in 2022, EC was the third-most common cancer diagnosis in women, at 29 new cases per 100,000 (compared to 27 new cases per 100,000 across the European Union) [5]. Given the aging world population, as well as the increased prevalence of diabetes and obesity, these figures are expected to grow further, which is why the early and correct diagnosis of EC is necessary [3]. By 2036, the global incidence of endometrial cancer among postmenopausal women aged 55 and older is projected to rise by 6.5%, while the mortality rate is expected to decline by 8.0%. Despite the anticipated reduction in mortality, the growing number of cases in this age group emphasizes the urgent need for a comprehensive screening and treatment strategy aimed at improving long-term survival outcomes [2]. Artificial intelligence is gaining attention in the wider field of gynecologic oncology, where it is being applied to improve diagnosis and treatment planning. While EC is a key focus, recent developments have also shown promise in other gynecologic cancers. Gynecologic cancers remain a major global health concern, accounting for approximately 15% of all new cancer cases and deaths among women in 2020. Cervical cancer was the most fatal gynecologic malignancy (7.7%), followed by ovarian (4.7%), uterine (2.2%), vulvar, and vaginal cancers. AI is being investigated as a tool for the early, non-invasive detection of ovarian cancer and for analyzing histopathological images to predict the molecular subtypes of endometrial tumors. By broadening the scope of AI applications, there is potential to enhance diagnostic precision and outcomes across the spectrum of gynecologic cancers [6].

The rise in EC incidence may be attributed to various factors, including the decreased use of approved hormone therapies, the increased use of compounded bioidentical hormone therapy, and the prevalence of obesity and diabetes [7]. An early diagnosis of EC is associated with a better survival rate, but such diagnosis has traditionally depended on invasive endometrial sampling. Using cytology and protein and DNA biomarkers in new, minimally invasive tests could revolutionize diagnostic processes and enable the monitoring of high-risk groups. The molecular classification of ECs has been shown to have therapeutic benefit, in addition to a prognostic influence, and it is increasingly being utilized to guide adjuvant treatment choices [3]. Current diagnostic modalities for endometrial cancer, such as transvaginal ultrasound, histopathological biopsy, and magnetic resonance imaging (MRI), each play a vital role in clinical assessment. Among these, MRI stands out for its superior soft-tissue contrast and use of non-ionizing radiation, making it highly valuable for pelvic imaging. However, its effectiveness is limited by sensitivity constraints, particularly in early disease detection. This challenge is typically addressed through the use of contrast agents that modulate T1 and T2 relaxation times of hydrogen protons, thereby enhancing image clarity. Despite these improvements, conventional MRI contrast agents face drawbacks, including potential toxicity and poor renal clearance. Integrating artificial intelligence into these imaging workflows can further enhance diagnostic accuracy by automating lesion detection, improving image interpretation, and supporting personalized risk stratification through data-driven insights [8].

Artificial intelligence offers a powerful tool for analyzing complex datasets, enabling the detection of nuanced patterns that may elude traditional human observation. By broadening the scope of AI applications, there is potential to enhance diagnostic precision and outcomes across the spectrum of gynecologic cancers. The primary objective of this review is to examine recent advancements in diagnostic approaches using AI for EC, with a particular focus on applications in histopathology, imaging, and multi-omics analysis. While AI has been explored in various oncologic contexts, its implementation in endometrial cancer diagnosis is still evolving. This review focuses on discussing the use of artificial intelligence and its implementations in diagnosing and treating EC, in addition to exploring possible disadvantages of artificial intelligence in this regard. It discusses the integration of AI tools in improving diagnostic precision and patient outcomes. The novelty of this study lies in its broad overview of the current AI methods used to diagnose endometrial cancer. It not only explains how these methods work, but also discusses how useful they are in medical settings. The review also points out important challenges, such as the need for more diverse data and better testing of these tools.

## 2. Materials and Methods

To gather the literature on AI-driven diagnostic innovations in EC, we conducted a comprehensive search of PubMed and Google Scholar databases, covering publications up to February 2025. The search terms included artificial intelligence, neural network, diagnostics, EC, histology, imaging, MRI, ultrasound, CT, multi-omics, and preoperative diagnostics. Only articles in the English language aligning with these terms were included. Studies were selected based on two criteria: they had to focus on the application of AI in EC and be full-text original research or review articles with complete sections (introduction, methods, results, and discussion). An initial screening of titles and abstracts was followed by full-text evaluations. Final article selection was refined through consensus among the authors to ensure the inclusion of the most pertinent sources, including 32 articles. Figure 1 summarizes the methodology.

## 3. Results

### 3.1. Histology

Histopathological analysis remains the gold standard for diagnosing EC, offering a structured and detailed method for identifying cancerous cells and assessing disease progression. Traditional manual pathology often falls short in evaluating subtle nuclear features and the architecture of tissues, limiting the ability to fully characterize the biological and clinical variations within EC. Integrating artificial intelligence into histopathology has the potential to revolutionize this process by capturing, analyzing, and classifying complex cellular patterns with greater speed and precision.

In a study by Zhang et al., a convolutional neural network (CNN) model was developed to automatically classify endometrial lesions using hysteroscopic images. Hysteroscopy is a widely used method for diagnosing endometrial lesions. The dataset consisted of 1851 images from 454 patients, covering various types of endometrial lesions. The model was trained with 6478 images and was tested on 250 images. The VGGNet-16 model achieved an overall accuracy of 80.8%, with strong sensitivity and specificity across different lesion types. When distinguishing between benign and premalignant/malignant lesions, the model showed 90.8% accuracy, 83.0% sensitivity, and 96.0% specificity. The model outperformed gynecologists in lesion-classification tasks, suggesting it could enhance the accuracy of the clinical diagnoses of endometrial lesions [9].

Fell et al. used artificial intelligence to classify endometrial biopsy whole-slide images (WSI) from digital pathology into three categories: “malignant”, “other or benign”, or “insufficient”. The AI model developed in the study aims to prioritize slides for pathologist review, potentially speeding up cancer diagnosis. The study used 2909 slides with annotated “malignant” and “other or benign” areas, training a convolutional neural network (CNN) to predict the likelihood of each patch being malignant. Heatmaps were generated to highlight malignant regions, and a final model was created to classify entire slides. The model achieved an accuracy of 90%, with 97% accuracy for malignant slides, making it effective for improving diagnosis outcomes [10].

In another study by Tahakashi, an artificial intelligence-based system was introduced that automatically detects areas affected by EC in hysteroscopic images. The study involved 177 patients, including those with normal endometrium, uterine myoma, endometrial polyps, atypical endometrial hyperplasia, and EC. The researchers used machine-learning techniques with three deep-neural-network models and developed a continuity-analysis method to improve diagnostic accuracy. The results showed that the standard method achieved an accuracy of around 80%, but using continuity analysis and combining the three models increased accuracy to 89%, and surpassed 90% (90.29%). The system demonstrated a sensitivity of 91.66% and specificity of 89.36%, suggesting it could enable more timely and accurate EC diagnosis in the future [11].

Li et al., in their study, developed and validated an artificial intelligence (AI) system to automatically recognize and diagnose endometrial cell clumps (ECCs) from cytology slides, addressing the global shortage of cytopathologists in EC (EC) screening. Using Li Brush sampling and liquid-based cytology, 113 patient samples (42 malignant, 71 benign) were collected, and 15,913 whole-slide images were processed. A U-Net segmentation network effectively isolated 39,000 ECC patches, which were then classified by a DenseNet201-based deep learning model. The system achieved 93.5% accuracy, 92.2% specificity, and 92.0% sensitivity on the test set, with the verification results perfectly matching expert pathologist labels. Moreover, when compared against other models such as VGG16, InceptionV3, ResNet, and SVM, DenseNet201 demonstrated a superior performance. These results highlight the potential of AI to significantly enhance diagnostic accuracy and efficiency in EC cytology, supporting its integration into clinical workflows to meet rising diagnostic demands [12].

A novel computer-aided diagnosis (CADx) system called HIENet was developed and introduced by Sun et al. The system was based on convolutional neural networks and attention mechanisms, and was designed to enhance the interpretation of histopathological images in the diagnosis of EC. Trained and validated on over 3500 hematoxylin and eosin (H&E) stained images, HIENet not only demonstrated strong classification performance but also provided interpretable visualizations to highlight relevant histological features for pathologists. In ten-fold cross-validation, the model achieved 76.91% accuracy for classifying four types of endometrial tissue (normal endometrium, endometrial polyp, endometrial hyperplasia, and endometrial adenocarcinoma) and showed excellent discrimination in the binary classification of malignancy, with an AUC of 0.9579, 81.04% sensitivity, and 94.78% specificity. External validation further confirmed its accuracy of 84.5% and an AUC of 0.9829, with perfect specificity (100%) (95% CI, 97.42–100.00%) and a high positive predictive value (100%) (95% CI, 92.29–100.00%). Notably, HIENet outperformed three expert pathologists and several classifiers based on convolutional neural networks, underscoring its potential as a valuable AI tool to augment diagnostic accuracy and efficiency in EC pathology [13].

Table 1 summarizes the reviewed literature on AI applications in histology for EC diagnostics.

### 3.2. Multi-Omics

Multi-omics refers to the simultaneous analysis of multiple layers of biological information, including the genome, transcriptome, proteome, metabolome, and epigenome. The role of multi-omics approaches in the diagnosis of EC is currently limited. These datasets provide valuable insights into the genetic, gene expression, protein, and metabolic alterations that characterize EC. By combining these datasets, researchers can identify novel molecular markers and therapeutic targets, which can enhance the accuracy of diagnosis and improve treatment strategies. The review by Boron et al. explains how next-generation sequencing (NGS), transcriptome profiling and proteomics can uncover critical genetic mutations, gene-expression changes, and protein alterations associated with the disease. Moreover, epigenetic profiling, such as DNA methylation analysis, can reveal further insights into the cancer’s progression. The integration of these diverse data types using advanced computational tools, including deep learning, has the potential to not only deepen our understanding of the biological mechanisms underlying EC but also to lead to more effective, personalized diagnostic and therapeutic approaches [14].

Using multi-omics and imaging-based analysis of endometrial carcinoma to improve patient stratification and treatment decision-making can offer a path toward more precise, personalized, and scalable approaches to EC diagnosis and treatment. This was explored by Dou et al. by conducting an in-depth molecular and proteogenomic analysis of 138 EC tumors and 20 enriched normal tissues using ten different omics platforms, including whole-genome sequencing (WGS), whole-exome sequencing (WES), various proteomics approaches, and RNA sequencing. The researchers uncovered key molecular features associated with EC progression and the treatment response. Notably, they developed a targeted peptide assay that accurately predicts antigen processing and presentation machinery (APM) activity, a potential biomarker for immunotherapy response. Additionally, AI deep learning models demonstrated strong a capability in predicting EC subtypes and key mutations directly from histopathology images, supporting the use of computational pathology for rapid, accessible diagnostics. The study also identified actionable molecular alterations, such as PIK3R1 in-frame insertions, which were linked to elevated AKT phosphorylation and heightened sensitivity to AKT inhibitors, suggesting a new biomarker for targeted therapy [15].

Hong et al., in their study, developed and evaluated Panoptes, a customized multi-resolution deep convolutional neural network. It is designed to predict not only histological subtypes but also molecular subtypes and gene mutations in endometrial carcinoma using digitized H&E-stained pathological images. The model demonstrated high accuracy and strong generalization on independent datasets, outperforming traditional methods in both predicting molecular subtypes and identifying 18 common gene mutations. The results indicate that, with further refinement, Panoptes could serve as a valuable tool in clinical practice, enabling pathologists to determine key molecular characteristics of endometrial carcinoma without the need for costly and time-consuming sequencing, thus potentially improving diagnostic efficiency and treatment selection [16].

Another study aimed to enhance the diagnostic approach to EC by integrating metabolomics and proteomics across multiple sample types, offering a more comprehensive understanding of the disease’s metabolic alterations. By analyzing endometrial tissue, urine, and intrauterine brushing samples from 44 EC patients and 43 controls, the researchers identified significant pathways related to EC, such as amino acid and nucleotide metabolism. Key metabolic changes observed in endometrial tissue were verified in both urine and intrauterine brushing samples, with notable differences in the trends between these sample types. This multi-omics approach highlighted the potential for non-invasive or minimally invasive diagnostic methods for EC, suggesting that altered metabolites and proteins could serve as valuable biomarkers for early detection [17].

In the United Kingdom, a study was conducted by Njoku et al. to develop a non-invasive diagnostic tool for EC through the use of AI-augmented proteomic analysis and focusing on protein biomarkers found in cervico-vaginal fluid and plasma. Using machine learning to analyze proteomic data from postmenopausal women with and without cancer, researchers identified a five-protein signature in cervico-vaginal fluid that could detect EC with high accuracy (AUC 0.95), sensitivity (91%), and specificity (86%). The cervico-vaginal fluid signature remained highly effective in detecting early-stage (stage I) cancers (AUC 0.92), highlighting its potential for early intervention. The findings suggest that proteins naturally shed by endometrial tumors into the lower genital tract can be harnessed through non-invasive fluid sampling, offering a safe, accurate, and scalable alternative to current invasive diagnostic methods [18].

In 2024, Volinsky-Fremond et al. introduced HECTOR, which is an AI-augmented, multimodal deep learning model designed to predict distant recurrence in EC using only routinely available clinical data, specifically H&E-stained whole-slide images and the tumor stage. The primary aim was to offer a more accessible and cost-effective alternative to the current gold standard, which combines clinicopathological risk factors with molecular profiling. While molecular testing improves prognostic accuracy, its high cost, turnaround time, and limited availability restrict widespread use. HECTOR was trained on a diverse dataset of over 2000 patients across eight cohorts and validated on both internal and external test sets. HECTOR demonstrated impressive prognostic performance, achieving C-indices of 0.789, 0.828, and 0.815 in internal and external test sets, respectively,—surpassing the predictive accuracy of conventional approaches. It effectively stratified patients into low-, intermediate-, and high-risk groups for distant recurrence, correlating with strikingly different 10-year recurrence-free survival outcomes (97.0%, 77.7%, and 58.1%, respectively). The model’s architecture combines self-supervised learning to extract morphological patterns from tumor images with the multimodal integration of tumor stage and image-based molecular classifications. Additionally, HECTOR was better at predicting who would benefit from adjuvant chemotherapy, addressing a critical clinical gap. Importantly, the model uses only standard pathology slides and basic clinical staging information, eliminating the need for complex and expensive molecular profiling, which is often inaccessible in routine clinical settings. The study also explored the biological correlates of the model’s risk stratification, revealing associations with both morphological features and genomic alterations, some of which may hold therapeutic significance [19].

Table 2 summarizes the reviewed literature on AI applications in multi-omics for EC diagnostics.

### 3.3. Imaging

Medical imaging and diagnostics have been revolutionized by artificial intelligence, particularly with machine learning and deep learning. Artificial intelligence is increasingly being applied to enhance diagnostic imaging in EC, offering new ways to detect patterns and biomarkers across multiple imaging modalities such as ultrasound, computed tomography (CT), and magnetic resonance imaging (MRI). These technologies have increased the diagnosis accuracy of numerous cancers, including EC. By analyzing large and complex datasets, AI systems can uncover subtle features that may be overlooked during routine evaluation, leading to earlier and more accurate detection of malignancy. In order to assist in early cancer detection and accurate diagnosis, artificial intelligence algorithms can analyze complex medical imaging data, such as ultrasound, CT, and MRI. These advanced AI-driven tools play a critical role in supporting the early detection of cancer and ensuring that diagnoses are as accurate as possible, ultimately improving patient outcomes and enabling more effective treatment planning [20]. Deep learning and machine learning approaches are being used to effectively diagnose, classify, and stage endometrial as well as cervical cancers. In a comprehensive review by Aparna et al., multiple studies and models have been mentioned that achieved high accuracy and even outperformed traditional screening methods like cytology and pap smears. Different techniques were used by the models, including texture analysis and radiomics. This review showed that deep learning and machine learning methods are becoming integral in cancer diagnostics as they enable more accurate, faster, and less invasive methods [21].

Ultrasound is an imaging technique used in the diagnostics of EC. In a study by the Mayo Clinic, researchers retrospectively analyzed data from 302 patients who underwent ultrasound and endometrial tests between 2016 and 2022. Physicians manually segmented ultrasound images, which were then used to train an AI-based automated segmentation model. Radiomic features such as image texture, shape, and intensity were extracted from the segmented regions to develop and evaluate multiple machine-learning classifiers. The top-performing classifier achieved high diagnostic accuracy, with an Area Under the Receiver Operating Characteristic curve AUC-ROC of 0.90 on validation and 0.88 on test data, along with strong sensitivity (0.87) and specificity (0.86) [22].

In a systematic review, the role of artificial intelligence (AI) in ultrasound imaging for gynecological oncology was explored. A total of 50 studies were reviewed, with 5 specifically addressing EC. These studies employed machine learning (ML) and deep learning (DL) models to predict malignancy, risk levels, and myometrial infiltration based on clinical and ultrasound variables. A total of 70.3% (26/37) evaluated AI models to distinguish benign from malignant lesions, with several achieving high accuracy (AUC up to 0.99). A total of 10.8% of the studies predicted tumor histology (e.g., benign vs. borderline vs. malignant) with AUCs up to 0.97. In EC, AI models demonstrated strong performance in predicting malignancy (AUCs 0.90–0.92), risk stratification, and myometrial invasion. For cervical cancer, models predicted lymph node metastasis (AUCs up to 0.82) [23].

Urushibara et al. conducted a study to compare the diagnostic performance of deep learning models, specifically convolutional neural networks with radiologists in diagnosing EC using MRI. The study included patients with either EC or non-cancerous lesions who underwent MRI between 2015 and 2020. In the first experiment, convolutional neural networks were trained using various image sequences from 204 cancer patients and 184 patients with non-cancerous lesions, and their diagnostic performance was tested on 97 images. The convolutional neural networks showed an area under the curve (AUC) between 0.88 and 0.95 for single and combined image sets, demonstrating diagnostic performance on par with the radiologists. In the second experiment, adding different types of images to the training set improved the diagnostic performance for some image sets, but the overall performance remained similar to the first experiment. The results indicated that convolutional neural networks can be highly effective in diagnosing EC using MRI and may offer diagnostic performance comparable to radiologists, with slight improvements when incorporating diverse image types into the training process [24].

A deep learning model was evaluated by Chen et al. in assessing myometrial invasion depth in EC using T2-weighted imaging (T2WI)-based MR imaging. A total of 530 patients with pathologically confirmed EC were included, and the MR images were divided into two groups based on the depth of myometrial invasion: deep (more than 50%) and shallow (less than 50%). The study employed the YOLOv3 algorithm to detect the lesion area in the MR images, and a classification deep learning model was used to identify the depth of myometrial invasion. The model demonstrated an average precision rate of 77.14% for sagittal images and 86.67% for coronal images. The classification model achieved an accuracy of 84.78%, sensitivity of 66.67%, and specificity of 87.50% in identifying deep myometrial invasion. When combined with radiologists, the model improved the diagnostic accuracy to 86.2%, with a sensitivity of 77.8% and a negative predictive value of 96.3%. However, the type of imaging used was not specified [25].

Tao et al., in their study, compared the MRI features of EC with surgical pathological results and explored the use of MRI in diagnosing EC. A total of 80 patients participated in the research and three different network models—shallow convolutional neural network, ResNet, and the optimized network—were used to analyze MRI images for recognition efficiency. The study utilized deep learning neural networks to perform quantitative and timed MRI analysis with simulated datasets, aiding in the development of single-scan MRI parameters. Of the 80 patients, 72 (90%) were diagnosed with stage I EC based on MRI, showing endometrial thickening and uneven enhancement [26]. Bourgioti et al. demonstrated that EC (EC) can be effectively identified on MRI using predictive algorithms, highlighting tumor size as an independent predictor of pelvic lymph node metastasis and myometrial invasion. Their model achieved 78% sensitivity, 92.7% specificity, and a positive predictive value of 90.5% [27]. In another study on MRI, Bereby-Kahane et al. assessed how well MRI-based texture analysis, tumor volume, tumor short axis, and the apparent diffusion coefficient predicted high grade tumors as well as lymphovascular space invasion in endometrial adenocarcinoma. TexRAD software (https://www.sciencedirect.com/science/article/pii/S2211568420300048, accessed on 26 April 2025) was used to assess texture data from 73 patients who had preoperative 1.5T MRIs. This model showed limitations of the model, with poor sensitivity and specificity for detecting high-grade malignancies (AUC = 0.64) and LVSI (AUC = 0.59). Tumor volume and short axis were significantly higher in high-grade and lymphovascular space invasion cases, while apparent diffusion coefficient values showed no correlation. The authors concluded that the ability of MRI-based texture analysis to predict EC high grade and LVSI is limited [28].

A study by Wang et al. investigated the clinical effectiveness of combining transvaginal ultrasound, magnetic resonance dispersion-weighted imaging (MRDWI), and multilayer spiral computed tomography in diagnosing early-stage EC. The dataset consisted of 100 cases, divided into a control group, which used conventional Doppler ultrasound, and an experimental group, which employed the combined imaging methods. The ultrasound images were preprocessed using a speckle-free adaptive Wiener filter and were segmented with fuzzy clustering. Feature extraction was performed using independent component analysis (ICA), and a deep VGG-16 AdaBoost hybrid classifier was applied to classify normal and abnormal images. The results showed that the diagnostic accuracy, specificity, sensitivity, and kappa coefficient were all significantly better in the experimental group, suggesting that the combined imaging methods, enhanced by AI techniques, provide superior clinical value [29].

Table 3 summarizes the reviewed literature on AI applications in imaging modalities for EC diagnostics.

### 3.4. Limitations

While AI-driven diagnostic tools offer transformative potential in improving the accuracy, efficiency, and personalization of EC diagnosis, several important limitations must be acknowledged. Many AI models are trained and validated on retrospective, single-institution datasets, raising concerns about their generalizability across broader, heterogeneous populations and healthcare systems [30]. One major concern is the “black box” nature of many deep learning models, which means their decision-making processes are not easily understood or explained. This lack of interpretability makes it difficult for clinicians to trust or validate the results, especially in high-stakes scenarios like cancer diagnosis [31]. Integrating AI tools into clinical practice also requires substantial digital infrastructure, clinician training, and standardized workflows, which may not be universally accessible, especially in resource-limited settings [32]. These limitations highlight the need for rigorous prospective validation, transparent model development, and better implementation strategies to ensure the safe and effective use of AI in EC diagnostics.

## 4. Conclusions

The integration of artificial intelligence (AI) into diagnostic workflows represents a significant advancement in the detection and management of endometrial cancer (EC). Across histopathology, imaging, and multi-omics domains, AI has demonstrated strong potential to enhance diagnostic accuracy, reduce subjectivity, and expedite decision-making. By leveraging standard clinical data and advanced algorithms, AI tools offer scalable, non-invasive solutions that complement or even approximate the performance of more complex molecular techniques. While further clinical validation is needed, current evidence supports the role of AI in enabling earlier diagnosis and improving risk stratification and more personalized treatment strategies, ultimately aiming to improve patient outcomes in EC care.

## Figures and Tables

**Figure 1 cancers-17-01810-f001:**
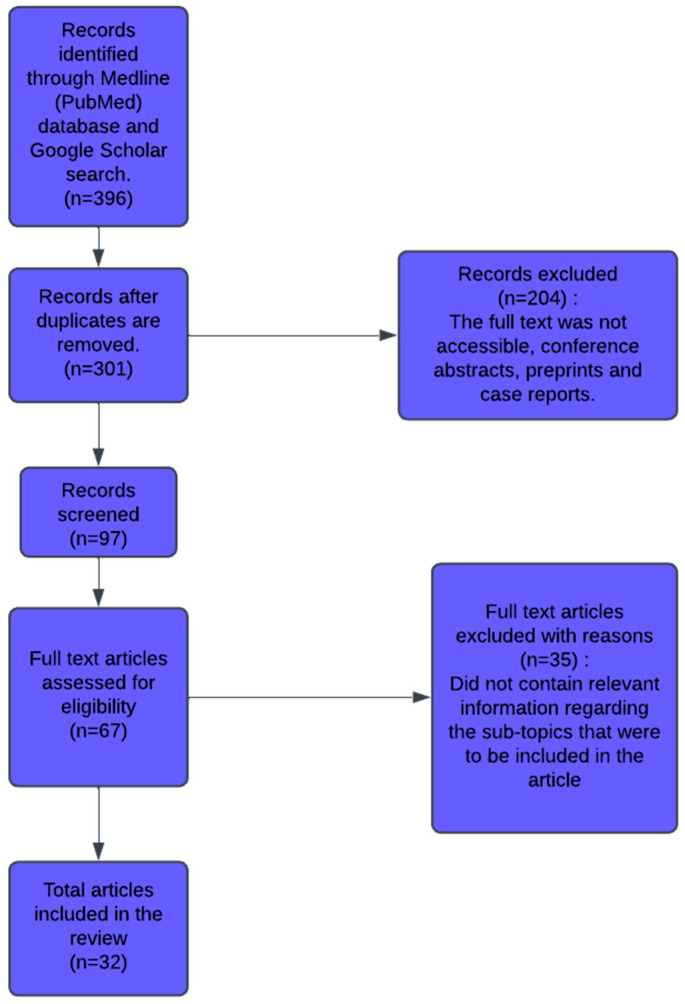
Methods of selecting the final articles included in this review.

**Table 1 cancers-17-01810-t001:** Summary of the reviewed literature on AI applications in histology for EC diagnostics.

Study	Dataset Size	AI Application	Main Results/Conclusions
Zhang et al. Journal of Translational Medicine (2021) [9]	1851 images from 454 patients; trained with 6478 images, tested on 250	CNN (VGGNet-16) for classifying endometrial lesions via hysteroscopic images	Achieved 80.8% accuracy overall; 90.8% accuracy distinguishing benign vs. malignant/premalignant lesions; outperformed gynecologists in lesion classification.
Fell et al. PLoS ONE (2023) [10]	2909 whole-slide images (WSIs) with annotated malignant and benign areas	CNN to classify endometrial biopsy WSIs into malignant, other/benign, or insufficient	Achieved 90% overall accuracy; 97% accuracy for malignant slides; potential to prioritize pathologist review and speed up cancer diagnosis.
Tahakashi et al. PLoS ONE (2021) [11]	177 patients with various endometrial conditions	Three deep-neural-network models with a continuity-analysis method for hysteroscopic image diagnosis	Combined model with continuity analysis reached 90.29% accuracy, 91.66% sensitivity, and 89.36% specificity; improved timely and accurate EC diagnosis.
Li et al. Cancers (2022) [12]	113 patient samples; 15,913 cytology slide images processed; 39,000 ECC patches	U-Net for segmentation and DenseNet201 for ECC classification from cytology slides	Achieved 93.5% accuracy, 92.2% specificity, and 92.0% sensitivity; outperformed other models and matched expert pathologists; supports integration of AI in cytological screening.
Sun et al. IEEE Journal of Biomedical and Health Informatics (2020) [13]	>3500 H&E-stained histopathological images; external validation conducted	CNN-based CADx system with attention mechanism for histological classification (HIENet)	Ten-fold CV accuracy of 76.91% (4-class); binary AUC of 0.9579; external validation accuracy 84.5%, AUC 0.9829; outperformed pathologists and existing CNN models in both interpretability and accuracy.

**Table 2 cancers-17-01810-t002:** Summary of the reviewed literature on AI applications in multi-omics for EC diagnostics.

Study	Dataset Size	AI Application	Main Results/Conclusions
Dou et al. Cancer Cell (2023) [15]	138 EC tumors + 20 normal tissues across 10 omics platforms	Deep learning to analyze histopathology images and predict EC subtypes and mutations	Identified new biomarkers (e.g., APM activity, PIK3R1 mutation); AI models effectively predicted EC subtypes and treatment-relevant mutations; supported computational pathology as diagnostic tool.
Hong et al. Cell Reports Medicine (2021) [16]	496 slides from 456 patients were included to form a mixed dataset	Multi-resolution CNN to predict histological and molecular subtypes, and gene mutations	Accurately identified EC subtypes and 18 common gene mutations; outperformed conventional methods; could replace some genomic testing with rapid image-based analysis.
Yi et al. Frontiers in Oncology (2022) [17]	44 EC patients + 43 controls	Multi-omics integration of metabolomics and proteomics	Identified metabolic alterations in tissues, urine, and brushings; highlighted potential for non-invasive diagnostic biomarkers for early EC detection.
Njoku et al. EBioMedicine (2024) [18]	53 symptomatic post-menopausal women with and 65 without endometrial cancer.	Machine learning on cervico-vaginal and plasma proteomic data	Identified five-protein signature with AUC 0.95, 91% sensitivity, 86% specificity; performed well even in stage I EC; supports non-invasive, fluid-based EC screening.
Volinsky-Fremond et al. Nature Medicine (2024) [19]	2000 patients across 8 cohorts	Multimodal deep learning model using H&E slides and tumor stage	Predicted distant recurrence with C-indices up to 0.828; stratified patients by recurrence risk; outperformed conventional risk stratification; requires only standard clinical data, increasing accessibility.

**Table 3 cancers-17-01810-t003:** Summary of the reviewed literature on AI applications in imaging for EC diagnostics.

Study	Dataset Size	Imaging Modality	AI Application	Main Results/Conclusions
Capasso et al. International Journal of Gynecological Cancer (2024) [22]	302 patients	Ultrasound	Radiomics + ML classifiers	Top classifier achieved AUC 0.90 (validation), 0.88 (test); sensitivity 0.87, specificity 0.86.
Moro et al. International Journal of Cancer (2024) [23]	50 studies (5 on EC)	Ultrasound	ML/DL for malignancy risk and invasion prediction	AUCs for EC prediction: 0.90–0.92; strong performance for malignancy risk and myometrial infiltration.
Urushibara et al. BMC Medical Imaging (2022) [24]	204 EC and 184 non-cancer patients	MRI	CNN (Deep Learning)	AUC: 0.88–0.95; performance comparable to radiologists; improved with diverse image training.
Chen et al. European Radiology (2020) [25]	530 EC patients	MRI	YOLOv3 + Deep Learning classifier	Accuracy: 84.78%; with radiologist: 86.2%; strong NPV (96.3%) for deep invasion detection.
Tao et al. Contrast Media & Molecular Imaging (2022) [26]	80 patients	MRI	Shallow CNN, ResNet, Optimized NN	90% of patients correctly identified as stage I EC; supports utility of MRI-based DL in diagnosis.
Bourgioti et al. Abdominal Radiology (2016) [27]	105 pateints	MRI	Predictive model	Sensitivity: 78%, Specificity: 92.7%, PPV: 90.5% for detecting pelvic lymph node metastasis.
Bereby-Kahane et al. Diagnostic and Interventional Imaging (2020) [28]	73 patients	MRI	Texture analysis (TexRAD)	Low AUC for high-grade tumor (0.64) and LVSI (0.59); limited predictive utility.
Wang et al. Journal of Oncology (2022) [29]	100 patients	Transvaginal ultrasound, MRDWI, CT	VGG-16 + AdaBoost	Combined imaging group had significantly better diagnostic performance across all metrics.

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
