# Peer review of "AI-Augmented Advances in the Diagnostic Approaches to Endometrial Cancer"

_cancers, 2025, doi:10.3390/cancers17111810_

Round 1

Reviewer 1 Report

Comments and Suggestions for Authors

  1. The abstract is informative and well-written.
  2. The introduction is very short; the authors should add recent studies applying AI in gynecologic oncology beyond endometrial cancer could be briefly mentioned to set a broader context.
  3. What I can see in the introduction is that the authors have provided old statistics to establish huge increments, including those from the UK and the USA. The authors should provide the current statistics.
  4. It would be more interesting if the authors provided statistics or an overview of Bulgaria & EU, as I can see the authors are from the same country. They can make an attempt.
  5. The 2nd paragraph should discuss the available diagnostic modalities, their limitations, and how AI can play the roles. The impact of medical imaging and its limitations have been discussed at https://doi.org/10.1021/acsbiomaterials.0c00409 . The authors are advised to follow and cite this article.
  6. In the 3rd paragraph, the authors should discuss the objective and novelty of this study.
  7. The Methods section lacks a systematic structure. Authors are advised to include a study selection flowchart and a number of included articles.
  8. It is suggested that authors add a table summarizing key studies (AI models, dataset size, performance metrics) for clarity and comparison.
  9. Authors are advised to add figures or schematic illustrations of AI workflows in histology/imaging, which would help with better understanding.
  10. Section 3.3: The authors should differentiate AI applied to different imaging modalities (ultrasound vs MRI vs CT) more distinctly in subheadings or tables for better understanding.
  11. Authors are suggested to add limitations of AI-driven diagnostic tools to take this work to the next level.

Author Response

Comment 1: The abstract is informative and well-written.

Response 1: Thank you, we appreciate the feedback.

Comment 2: The introduction is very short; the authors should add recent studies applying AI in gynecologic oncology beyond endometrial cancer could be briefly mentioned to set a broader context.

Response 2: Agreed. As suggested, we have expanded the introduction to include recent studies highlighting the application of artificial intelligence in gynecologic oncology beyond endometrial cancer.  This addition provides a broader context for the relevance and potential of AI in the diagnosis and management of various gynecologic malignancies. We have also incorporated global statistics to further emphasize the significance of this research area.

Comment 3: What I can see in the introduction is that the authors have provided old statistics to establish huge increments, including those from the UK and the USA. The authors should provide the current statistics.

Response 3: Thank you for pointing this out. We agree with this comment and therefore have removed the older statistics and included current global statistics. 

Comment 4: It would be more interesting if the authors provided statistics or an overview of Bulgaria & EU, as I can see the authors are from the same country. They can make an attempt.

Response 4: This is a great point. We have added recent statistics from Bulgaria and Europe. 

Comment 5: The 2nd paragraph should discuss the available diagnostic modalities, their limitations, and how AI can play the roles. The impact of medical imaging and its limitations have been discussed at https://doi.org/10.1021/acsbiomaterials.0c00409 . The authors are advised to follow and cite this article.

Response 5: We appreciate the reviewer’s insightful suggestion. In response, we have revised the second paragraph to include a discussion of current diagnostic modalities used in endometrial cancer, their limitations, and the potential roles of artificial intelligence in addressing these challenges. Specifically, we have elaborated on the strengths and limitations of imaging techniques, with a focus on magnetic resonance imaging (MRI), and incorporated content from the recommended article. 

Comment 6: In the 3rd paragraph, the authors should discuss the objective and novelty of this study.

Response 6: Thank you for your valuable feedback. As suggested, we have revised the third paragraph to outline the objective and novelty of our study. 

Comment 7: The Methods section lacks a systematic structure. Authors are advised to include a study selection flowchart and a number of included articles.

Response 7: Thank you for the suggestion. We have added a flowchart for the methods section and a number of included articles.

Comment 8: It is suggested that authors add a table summarizing key studies (AI models, dataset size, performance metrics) for clarity and comparison.

Response 8: We thank the reviewer for their insightful suggestion. In response, we have added 3 comprehensive tables summarizing the key studies referenced in our manuscript for clarity and comparison. 

Comment 9: Authors are advised to add figures or schematic illustrations of AI workflows in histology/imaging, which would help with better understanding.

Response 9: We thank the reviewer for their insightful suggestion. To better understand AI workflows and their independent use cases, we included 3 tables which summarize AI tools and their relative applications.

Comment 10: Section 3.3: The authors should differentiate AI applied to different imaging modalities (ultrasound vs MRI vs CT) more distinctly in subheadings or tables for better understanding.

Response 10: We agree the imaging modalities should be listed more distinctly. We have differentiated AI and the imaging modalities in Table 3 for a better understanding. Thank you for pointing this out!

Comment 11: Authors are suggested to add limitations of AI-driven diagnostic tools to take this work to the next level.

Response 11: Thank you for your suggestion. We have added a section on limitations to discuss the next steps towards research in AI diagnostics.

Reviewer 2 Report

Comments and Suggestions for Authors

I congratulate the authors on the concept of artificial intelligence in the diagnosis of endometrial pathology.

In the methodology section, in my opinion, there is no graphical representation of the method of selecting articles chosen for this analysis. I suggest creating an appropriate flow chart.

The conclusions section is too extensive to make it difficult for a potential reader to evaluate the work. In my opinion, some of the information contained in the conclusions section should be moved to the results section. - In this way, the conclusions section will be more condensed and therefore more understandable for the reader.

Author Response

Comment 1: I congratulate the authors on the concept of artificial intelligence in the diagnosis of endometrial pathology.

Response 1: Thank you, we appreciate it!

Comment 2: In the methodology section, in my opinion, there is no graphical representation of the method of selecting articles chosen for this analysis. I suggest creating an appropriate flow chart.

Response 2: Thank you for the suggestion. We have added a flowchart for the methods section.

Comment 3: The conclusions section is too extensive to make it difficult for a potential reader to evaluate the work. In my opinion, some of the information contained in the conclusions section should be moved to the results section. - In this way, the conclusions section will be more condensed and therefore more understandable for the reader.

Response 3 : We agree that the original conclusion was overly detailed and included content more appropriate for the Results section. In response, we have revised the Conclusions section to make it more concise and focused on key takeaways, while integrating the more specific findings into the Results section.

Round 2

Reviewer 1 Report

Comments and Suggestions for Authors

The revised version may be accepted for publication as is, with the additional advice for the authors to change the color scheme of Figure 1.